# Changes in extreme precipitation patterns over the Greater Antilles and teleconnection with large-scale sea surface temperature

Carlo Destouches[1,2], Arona Diedhiou[2], Sandrine Anquetin[2], Benoit Hingray[2], Armand Pierre[1], Dominique Boisson[1], and Adermis Joseph[1]

[1]Université d'Etat d'Haïti, Faculté des Sciences, LMI CARIBACT, URGéo, Port-au-Prince, Haïti
[2]Univ. Grenoble Alpes, IRD, CNRS, Grenoble INP, IGE, F-38000 Grenoble, France

**Correspondence:** Carlo Destouches (carlo.destouches@univ-grenoble-alpes.fr)

**Abstract.** This study examines changes in extreme precipitation over the Greater Antilles and their correlation with large-scale sea surface temperature (SST) for the period 1985 to 2015. The data used for this study were derived from two satellite products: Climate Hazards Group InfraRed Precipitation (CHIRPS) and NOAA DOISST (Daily Optimum Interpolation Sea Surface Temperature version 2.1) with resolutions of 5 km and 25 km, respectively. Then, change in the characteristics of
six(6) extreme precipitation indices defined by the WMO ETCCDI (World Meteorological Organization Expert Team on Climate Change Detection and Indices) is analyzed, and Spearman's correlation coefficient has been used and evaluated by t-test to investigate the influence of a few large-scale SST indices: (i) Caribbean Sea Surface Temperature (SST-CAR); (ii) Tropical South Atlantic (TSA); (iii) Southern Oscillation Index (SOI); (iv) North Atlantic Oscillation Index (NAO). The results show that at the regional scale, +NAO contributes significantly to a decrease in heavy precipitation (R95p), daily precipitation inten-
sity (SDII), and total precipitation (PRCPTOT), whereas +TSA is associated with a significant increase in daily precipitation intensity (SDII). At an island scale, in Puerto Rico and southern Cuba, the positive phase of +TSA, +SOI, and +SST-CAR is associated with an increase in daily precipitation intensity (SDII) and heavy precipitation (R95p). However, in Jamaica and northern Haiti, the positive phases of +SST-CAR and +TSA are also associated with increased indices (SDII, R95p). In addition, the SST warming of the Caribbean Sea surface temperature and the positive phase of the Southern Oscillation (+SOI) is
associated with a significant increase in the number of rainy days (RR1) and the maximum duration of consecutive rainy days (CWD) over the Dominican Republic and in southern Haiti.

Keywords: Caribbean region; Greater Caribbean; Extreme precipitation; Climate variability; Sea surface temperature

## 1   Introduction

Over the past three decades, the climatic hazards to which the Caribbean Basin has been exposed include recurrent cyclonic and hydrometeorological hazards, characterized by increasing intensities (Joseph, 2006). The economic cost of 250 storms and floods over 40 years (1970 to 2009) for 12 Caribbean countries amounted to US19.7 billion in 2010, representing an annual average of 1% of gross domestic product (GDP) (Burgess et al., 2018). The most dreadful damage caused by these

hydroclimatic events includes George in 1998, with 1,000 victims in the Dominican Republic and losses estimated at 14% of GDP, equivalent to approximately half the exports made that year (Naciones Unidas, Comision Economica para America Latina y el Caribe, 1988); Matthew in Haiti (October 2016), with over 500 dead, 128 missing, 439 injured and 2.1 million people affected, including 895,000 children(De Giogi et al., 2021). Also, Hurricane Dorian caused property damage estimated at 2.5 billion USD when it came to rest over the Bahamas as a Category 5 storm in September 2019, rendering nearly 3,000 homes uninhabitable and causing extensive damage to hospitals, schools, and fisheries (Panamerican Health Organisation, 2019). A severe drought episode affected the island of the Caribbean from October 2019 to mid-2020, causing water shortages, bushfires, and agricultural losses. In Saint Vincent and the Grenadines, the 2020 drought was considered the worst of the 50 years (Nurse, 2020). The Inter-American Development Bank predicts that the Caribbean could face climate-related losses of over $22 billion per year by 2050 (Inter-American Development Bank, 2014).

In response to the climate extremes that are further weakening the island states of the Caribbean region, already in a situation of extreme socio-economic precariousness, several studies have been carried out in parallel to understand the associated physical processes and anticipate the evolution of these extreme climatic events. Research into the Caribbean climate goes back to the second half of the twentieth century and has focused mainly on rainfall patterns (Curtis et al., 2008), as well as on the overall description of rainy seasons (Griffiths et al., 1982).

A more detailed study of the climate of the Caribbean was performed in 2001 and 2002 using indices derived from daily data to detect climate change (Peterson et al., 2001; Frich et al., 2002). This approach, which uses indices defined by the World Meteorological Organization's group of experts to characterize precipitation and temperature extremes, has enabled several studies to examine the state of climate extremes over the Caribbean (Stephenson et al., 2014; McLean et al., 2015). The results of these previous assessments agree that the frequency and intensity of climate extremes(heavy rainfall, drought spell, wet spell ) over the Caribbean have increased over the last 30 years (Stephenson et al., 2014; Peterson et al., 2002a; Beharry et al., 2015; Dookie et al., 2019), and will continue to do so until the end of the century(Taylor et al., 2018; Vichot-Llano et al., 2021; Hall et al., 2013; Almazroui et al., 2021; McLean et al., 2015).

Climate teleconnections, the remote forcing of a region far from the source of disturbance, whether simultaneous or time-lagged (Mariami et al., 2018; Rodrigues et al., 2021), are generally derived from variations in sea surface temperature (SST) or atmospheric pressure at seasonal to interdecadal scales. Several of these have been shown to play a major role in modifying global weather patterns (Hurrell et al., 1995; Martens et al., 2018).

Previous studies have also shown the effect of east-west gradients in SST anomalies in the tropical Pacific and Atlantic on precipitation in the Caribbean, with a tendency for a warm Atlantic and a cold Pacific to favor precipitation in the Caribbean (Taylor et al., 2002a; Gimeno et al., 2011). Studies (Enfiel et al., 2001; IPCC, 2007) also found that the monthly AMO (Atlantic Multidecadal Oscillation) index is an SST signal in the North Atlantic that influences the decadal-scale variability in precipitation. In addition, Peterson et al. (2022) analyzed the link between SST, temperature, and precipitation extremes over the Caribbean using ground-based observations. They showed that the extreme precipitation index (SDII) averaged over the Caribbean has a strong correlation with SST over the Caribbean and the entire tropical North Atlantic Ocean. The work of Stephenson et al. (2014) examined the influence of the Atlantic Multidecadal Oscillation (AMO) on extreme precipitation from

a ground-based observation network in the Caribbean. These results show that the AMO influences the variability of extreme temperature and precipitation events. However, considering that the effects of teleconnections caused by large-scale SSTs on weather are expected to become more extreme in the future due to climate change (Mariami et al., 2018), further research is needed on other SST indices such as NAO, SOI, TSA, SST-Car, for which no in-depth studies have been carried out.

In this context, this study aimed to examine the remote impact of the tropical Pacific Ocean, Atlantic Ocean, and the Caribbean Sea on observed changes in the tropical islands of the Greater Antilles, particularly the links between extreme precipitation indices and large-scale sea surface temperature indices. This paper is organized into five sections: Section 1 presents the study area and the associated climatology. Sections 2 and 3 describe the spatio-temporal variability of extreme precipitation indices at regional and local scales and the influence of SST indices on extreme precipitation. The last sections (4-5) present the discussion and conclusion.

## 2   Study Area and Data

### 2.1   Study area

The Greater Antilles is a region between North and South America made up of four islands bordered by the Caribbean Sea to the south and the Atlantic Ocean to the east (Fig.1). These islands include Cuba, Hispaniola, Jamaica, and Puerto Rico. They have a monthly rainfall cycle characterized by two peaks: the first in May and the second between September and October (Giamini et al., 2000). The climatology of monthly rainfall in the Greater Antilles is strongly influenced by the subtropical North Atlantic anticyclone (Davis et al.,1997), low-level jet (CLLJ), characterized by two peaks; the first in January and the second in July (Cook and Vizy, 2010). This jet plays a key role in transporting moisture to the Caribbean (Mo et al., 2005). They were also influenced by the intertropical convergence zone (ITCZ) (Hastenrath, 2002), with maximum precipitation in May (sup.fig.1b). Heavy autumn rainfall in the Greater Antilles (supl.fig.1d) is generally associated with North Atlantic tropical cyclones, 85% of which are of high intensity and originate from African easterlies (Agudelo et al., 2011; Thorncroft and Hoges, 2001) under warm Atlantic basin conditions. The spatial distribution of the total annual precipitation in the Greater Antilles, particularly on the islands, is not homogeneous due to the complexity of topography (Moron et al., 2015, Cantet, 2007). Precipitation is relatively high (2,000-24,000< mm/year) at higher altitudes and in wind-exposed areas (supl.fig.1b). In contrast, annual precipitation can reach 500 mm/year in leeward areas (Daly et al., 2003).

### 2.1.1   Satellite data

This study was conducted using two satellite datasets: NOAA DOISST Sea surface temperature data and CHIRPSv2 data. The CHIRPSv2 data (Climate Hazards Group Infrared Precipitation with Stations data version 2) are quasi-global daily precipitation data (50S-50N) with a resolution of 0.050, available over a period from 1981 to 2022(Funk et al., 2015). Based on the techniques used by NOAA for estimating precipitation in the thermal infrared (Love et al., 2004), the CHIRPSv2 database was built from precipitation estimates based on cold cloud duration observations, and a fusion incorporating monthly CHPClim

(Funk et al., 2015a) (Climate Hazards Group Precipitation Climatology (CHPClim) precipitation data, and in situ data from ground observation networks. TRMM 3B42v7 (Tropical Rainfall Measuring Mission Multi-Satellite Precipitation Analysis) satellite products were also used to calibrate and reduce the bias in the estimates. The results of global and regional validation studies showed that CHIRPSv2 can be used to quantify the hydrological impacts of decreasing rainfall and increasing air temperatures in the Greater Horn of Africa (Funk et al., 2015). In addition, the performance of CHIRPSv2, evaluated over certain regions of the Americas, has demonstrated its ability to reproduce the mean climate as well as its capacity to estimate extreme precipitation events (Rivera et al., 2019). Furthermore, in Colombia, the best results were obtained on a daily and monthly scale over the Magdalena River Basin (Baez-Villanueva et al., 2018). CHIRPSv2 data are suitable for our study, as they perform well in the Caribbean (Centella-Artolla al., 2020), and in the study by Bathelemy et al.(2022), it was shown that Chips perform well in estimating heavy precipitation based on the 90th percentile(Bathelemy et al., 2022).

Sea surface temperatures (SST) are very important for monitoring and assessing climate change (IPCC, 2013). They can be derived either from observations from floating or moored buoys (Smith et al., 1996), from satellite observations (Merchant et al., 2014), or from a mixture (in situ + satellite) (HadSST, Rayner et al., 2003) and (DOSST, Reynold et al., 2007). In this study, NOAA DOISST (Daily Optimum Interpolation Sea Surface Temperature version 2.1) data were used; these are also daily sea surface temperature data derived from a combination of in situ sea surface temperature (SST) data obtained from ships and buoys and sea surface temperatures obtained from the Advanced Very High-Resolution Radiometer (AVHRR)(Reynold et al., 2007; Huang et al., 2021). This satellite product is the result of a global file of 0.250-degree grid points, available over the period 1981-2020. In addition, it has been widely used for climate assessment and monitoring, notably as part of the reanalysis of the NOAA/NCEP climate prediction system (Saha et al., 2010). Work by Huang et al.(2021a) has revealed that NOAA DOISST performs well in terms of bias compared with buoy and Argo observations, as well as with the eight SST products.

## 3 Methodology

The World Meteorological Organization's Expert Team on Climate Change Detection and Indices (ETCCDI) has defined 27 indices to characterize extreme precipitation and temperature events in terms of frequency, amplitude, and duration (Peterson et al., 2001). Although the proposed method includes numerous indices based on percentiles, with thresholds set to assess extremes that generally occur a few times a year and not necessarily high-impact events, it has paved the way for numerous research projects in the Caribbean (Stephenson et al., 2014; McLean et al., 2015). Six extreme precipitation indices (see Table 1 for details) were calculated: total annual precipitation (PRCPTOT), number of rainy days (RR1), intensity of rain events (SDII), and heavy precipitation (R95p), calculated with a threshold corresponding to the 95th percentile of the daily precipitation distribution, maximum number of consecutive wet days (CWD), and maximum number of consecutive dry days (CDD).

The spatiotemporal evolution of extreme precipitation in the Greater Antilles was investigated by analyzing the interannual variability of extreme precipitation index anomalies over a long period (1985-2015) and the change in percentage variations in

extreme precipitation indices at decadal timescales. To characterize the percentage variations, we chose the (Pij) index, which is already used in the study by An et al. (2023), whose equation is presented hereafter.

$$P_{ij} = \left( \frac{P_{i(j+1)}}{P_{ij}} - 1 \right) \times 100 \tag{1}$$

125    Where Pij is the average extreme precipitation index for the j-th decade at the i-th location, and Pi(j+1) is the average extreme precipitation index for the (j+1)-th decade at the i-th location.

Previous studies have shown that precipitation in the Caribbean, particularly in the Greater Antilles, is influenced by the surface temperature of the Atlantic Ocean and tropical Pacific (Gimeno et al., 2011; Enfiel et al., 2001). Thus, to investigate
the impact of these basins on extreme precipitation over indices, we selected four large-scale SST indices, namely the Southern Oscillation Index (SOI), the North Atlantic Oscillation (NAO) (Jones et al., 1997), the Tropical South Atlantic Anomaly Index (TSA) and the Caribbean Sea Surface Temperature Anomaly Index (SST-CAR). Details of these indices can be consulted online: https://psl.noaa.gov/data/climateindices.

Analysis of the relationship between two variables is often of great interest for data analysis in research. It generally consists
in characterizing the form and intensity of the link (relationship) between variables using a correlation coefficient. For two variables, X and Y, this coefficient is interpreted as: i)linear linkage, the correlation coefficient is positive when X and Y values change in the same direction, that is, an increase in X leads to an increase in Y; ii)linear linkage, the correlation coefficient is negative when X and Y values change in the opposite direction, that is, an increase in X leads to a decrease in Y (or vice versa); iii) non-linear monotonic linkage, the correlation coefficient is positive when X and Y change in the same direction as in (i),
but with a small slope (Lewis-Beck, 1995; Sheskin, 2007). In the literature, Pearson's and Spearman's correlation coefficients are often the most widely used to measure the strength or degree of linkage between two variables. In this study, we used Spearman's non-parametric rank correlation coefficient (rho) to assess the interannual link between extreme precipitation over indices and global SSTs indices. This non-parametric method was chosen because it does not require a normal distribution for the variables. Also, Spearman outperforms Pearson's linear coefficient in the case of outliers. On the other hand, linear
trends can be detected using Pearson or Spearman tests, but the latter is preferable for monotonic non-linear relationships (Gauthier, 2001; Von Storch and Zwiers, 1999). Spearman's correlation has already been used in several studies to assess the links between teleconnection patterns and precipitation. (Ríos-Cornejo et al., 2015; Khadgarai et al., 2021).

Spearman's correlation coefficient is calculated using the following equation:

$$r_{spearman} = \frac{\sum_{i=1}^{n}(R_i - \overline{R})(S_i - \overline{S})}{\sqrt{\sum_{i}^{n}(R_{i=1} - \overline{R})^2 \sum_{j=1}^{n}(R_{i=1} - \overline{R})}} \tag{2}$$

Where $r_{spearman}$ is the correlation coefficient, Ri=rang (Xi), Si=rang (Yi) are respectively the data ranks of Variables X and Y (X: Extreme precipitation index, Y: Large-scale SST index).

To test the significance of the relationship, whether the two variables are correlated or not, we use the t-test for a threshold of 0.05 or less. This involves testing the two hypotheses ($H_0$ and $H_1$) based on the value of t to deduce the probability of observing a result that deviates as much as expected from the correlation. The formula for calculating the value of t using Spearman's correlation is as follows:

$$t_{n-2} = \frac{r_{spearman}}{\sqrt{1 - r_{spearman}^2} \times \sqrt{n-2}} \tag{3}$$

Where:

n-2: degrees of freedom.

n: sample size

## 4 Results

### 4.1 Changes in precipitation extreme indices

Interannual changes in extreme precipitation indices over the Greater Antilles are shown in Fig. 2. As shown in Fig. 2a, the period from 1985 to 1994 was generally marked by a decline in total annual precipitation. This decline was associated with a decrease in average rainfall intensity (Fig. 2c) and a reduction in the length of wet and dry spells (Fig. 2e, 2f). On the other hand, the period from 1995 to 2004 was mainly characterized by a decrease in the number of rainy days (fig.2b), associated with an increase in the average intensity of precipitation (fig. 2c) and in the contribution of heavy precipitation (fig. 2d). Furthermore, during the period 2005-2015, an increase in the contribution of heavy precipitation was observed until 2012(fig.2d). This was associated with an increase in the number of rainy days(fig.2b), an increase in the average intensity of precipitation(fig.2c) and in the length of wet episodes(fig.2e).

Fig. 3 shows the annual change in percentage between two consecutive decades of precipitation indices over the Greater Antilles (1985-2015). As shown in fig. 3(a, d2), there was an increase in total annual precipitation (PRCPTOT) in southeastern Cuba. This was associated with an increase in the number of rainy days (RR1) (fig.3(b, d2)) and the average intensity of precipitation per rainy day (SDII) (fig.3(c, d2)). These results were also observed in Puerto Rico, with an increase in total annual precipitation associated with an increase in RR1 (fig.3(a, d2), 3(b, d2)). In addition, a decrease in total annual precipitation (PRCPTOT) was observed on the island of Hispaniola (Dominican Republic and Haiti, except for the southern part ) (fig.3(a, d2)). This was associated with a decrease in the average rainfall intensity per wet day (SDII) (fig.3(c, d1)). This decrease in the SDII was also recorded in Puerto Rico (fig.3(c, d2)). For heavy precipitation (R95p), as shown in fig.3(d, d2), an increase was observed in the southeastern part of Cuba, whereas the whole island (Cuba) was affected in general by a decrease in wet sequences (CWD) (fig.3(e, d2)) and dry sequences (CDD) (fig.3(f, d2)). A decrease in heavy precipitation (R95p) was observed in the central and western regions of Haiti (fig.3(d, d2)). This was accompanied by an increase in wet sequences (CWD) over Haiti (fig.3(e, d2)). The Dominican Republic was also affected by this increase in wet sequences (CWD) (fig.3(e, d2)).

Variations in extreme precipitation indices under the influence of variables such as NAO, SOI, TSA, and SST-CAR were analyzed over the Greater Antilles. The influences of large-scale variables were classified as positive, negative, positive, significant, negative, or significant, as shown in figure 4. The results obtained by taking the intersections of the table in fig. 4 presented show the values of the correlation coefficient (with its significance *) between the extreme precipitation indices (PRCPTOT, RR1, SDII, R95p, CWD, and CDD) and the influencing variables (NAO, SOI, TSA, and SST-CAR). Thus, the table in fig. 4 shows that NAO has a negative effect on all extremes, while the other SST-CAR are positive, except for the number of rainy days (RR1) and the number of consecutive rainy days (CWD). However, the positive phase of the +TSA index had a positive and significant effect on the average rainfall intensity per wet day (SDII), for which a correlation coefficient of 0.37 was obtained. Similarly, with the ONA index, a negative and significant effect ($P<0.05$) was observed on total annual precipitation (PRCPTOT), average precipitation intensity (SDII), and heavy precipitation (R95p), for which correlation coefficients of 0.49, 0.40, and 0.47, respectively, were obtained.

At a local scale, the results show that teleconnections have had positive and significant effects on extreme precipitation indices over the last 30 years in the countries of the Caribbean region, particularly in the Greater Antilles. The spatial extent of significance is indicated by the symbols (*). Thus, the double symbol (**) represents regions with a significant surface area greater than 50% of the surface area, while the symbol (*) is used for a significant surface area less than 50%Thus, a double symbol (**) indicates that the effect extends over a wide area, whereas the effect is limited to that bearing a single symbol (*). The figures (with a threshold at $p \leq 0.05$; fig. 5, 6, 7, 8) show the regions or countries over which positive and significant effects were observed for the Greater Antilles.

Fig. 5 and suppl. Table 1d shows the effect of the TSA index on extreme precipitation indices (PRCPTOT, RR1, SDII, R95p, CWD, and CDD) in the Greater Antilles. The results show that South Atlantic tropical warming, corresponding to the positive phase of the +TSA index, has a positive and significant effect (**) on the total annual precipitation (PRCPTOT) and heavy precipitation (R95p) in Puerto Rico (fig.5a, 5b; suppl. table1d). In Jamaica, it was also associated with an increase in total annual precipitation (PRPCPTOT) and heavy precipitation (R95p) (fig.5a, 5b; suppl. table1d). In Haiti, more specifically in the northern part, the increase in the average daily rainfall intensity (SDII) was also associated with +TSA warming (fig.5e; suppl. table1d). In northwest Cuba, this positive phase of +TSA also had a positive and significant effect (**) on mean rainfall intensity per day (SDII) (fig.5e; suppl. table1d). Conversely, in southeastern Cuba, a positive effect was observed for heavy rainfall (R95p) (fig.5b; suppl. table1d).

Fig. 6 and suppl. Table 1a shows the effects of warming of the Caribbean Sea surface temperature (SST-Car anomaly, averaged over 14-16N, 65–85 °W) on extreme precipitation in the Greater Antilles. In southern Haiti, an increase in total annual precipitation (PRCPTOT) and the number of rainy days (RR1) was observed as the SST-Car warmed (fig.5a, 5c; suppl. table1a). In the same region, this increase was also observed for heavy rainfall (R95p) (fig.6b; Suppl. table1a). In eastern Haiti, particularly in Santo Domingo, the positive phase of + SST-Car is associated with an increase in the duration of wet sequences (CWD) and the number of rainy days (RR1) (fig.6c, 6d; suppl. table1a). In southeastern Cuba, especially on the Caribbean coast, warming of SST-Car is associated with an increase in total annual precipitation (PRCPTOT) and heavy precipitation

(R95p) (fig.6a, 6b; suppl. table1a). An increase in heavy precipitation (R95p) during the positive phase of + SST-Car was also observed in Puerto Rico (fig.6c; Suppl. table1a).

Fig. 7 and suppl. Table 1c shows the results of the effect of SOI on extreme precipitation indices (PRCPTOT, RR1, SDII, R95p, CWD, and CDD) in the Caribbean region, particularly on the islands forming the Greater Antilles. In Puerto Rico, as shown (fig.7a, 7b; suppl. table1c), the positive phase of the +SOI index was associated with an increase in total annual precipitation (PRCPTOT) and heavy precipitation (R95p). In Haiti, specifically in the south, this positive phase of the +SOI index was associated with an increase in the number of rainy days (RR1), including the duration of wet sequences (CWD) (fig.7c, 7d; suppl. table1c). In Santo Domingo, this is associated with an increase in the duration of wet sequences (CWD) (fig.7d; suppl. table1c). In southeastern Cuba, this is associated with an increase in the average intensity of precipitation per rainy day (SDII) and heavy precipitation (R95p) (fig.7e, 7b; suppl. table1c).

Fig. 8 and suppl. Table 1b shows the results of the effect of the NAO index on extreme precipitation indices (PRCPTOT, RR1, SDII, R95p, CWD, and CDD) in the Greater Antilles. The condensed results for different phases of the NAO index are presented in Supplementary Table 1b. In contrast to the results for the other large-scale SST indices, the positive phase of the +NAO index was only associated with an increase in the number of rainy days (RR1) over Cuba. This increase was greater for coasts facing the Caribbean Sea (fig.8c; Suppl. Table).

## 5    Discussion

The results of this study showed that extreme precipitation over the period 1985-2015 was influenced by four large-scale sea surface temperature indices (NAO, SOI, TSA, SST-Car). On the other hand, on a local scale, notably over a few regions, the effects of this influence on certain precipitation indices were statistically significant, while on other precipitation indices, non-significant effects were observed. To shed more light on these results, we discuss the following points: i) Impact of sea surface pressure anomalies (NAO, SOI); ii) Impact of sea surface temperature anomalies (TSA, SST-Car).

### 5.1    Impact of sea surface pressure anomalies (NAO, SOI)

In this study, the influence of the Atlantic Ocean over the Greater Antilles is assessed using two large-scale SST indices (NAO, Hurrell (2003)) and TSA. The phases (positive and negative) of the NAO index have been shown to affect circulation in the Northern Hemisphere (Thompson et al., 2000). Also, this index is a measure of the meridional pressure gradient between the NASH and the Icelandic low (Visbeck et al., 2001). Nevertheless, in this study, the results show that the NAO has a negative effect on all extreme precipitation over indices in the region (fig.4). In other words, the fluctuation of the NAO index and extreme precipitation over indices change in opposite phases, i.e. the negative (or positive) phase of the NAO is linked to the positive (or negative) phase of the precipitation indices. However, on a local scale (Fig. 8), notably in Cuba and southern Haiti, the positive effect of NAO+, corresponding to a weakening of the subtropical anticyclone (Wallace et al., 1981; North Atlantic Oscillation, 2023) and leading to very wet conditions in the Caribbean, particularly in the Greater Antilles (Giannini et al.

,2000; Mo et al., 2005), is associated with an increase in the number of rainy days (RR1). These results are consistent with those of Jury et al. (2007), for whom the NAO exerts a certain influence on rainfall in the southeastern Caribbean.

In the case of the SOI index (fig.7), studies have already shown that ENSO influences precipitation patterns in several regions, notably in South and North America (Ropelewski et al., 1987). However, studies (Giannini et al., 2000; Giannini et al., 2001a; Giannini et al., 2001b; Rodriguez-Vera et al., 2019) have shown that, on an interannual scale, ENSO is one of the most important factors influencing precipitation in the Caribbean. These results are consistent with those of our study. The influence of ENSO was assessed on extreme precipitation indices (figure 7) using the SOI index. For example, the positive

phase of the +SOI index (sign of La Niña), characterized by abnormally cold ocean waters in the eastern tropical Pacific, led to an increase in total annual precipitation (PRCPTOT) and heavy precipitation (R95p) in Puerto Rico (fig. 7a, 7b). It has also led to an increase in the number of rainy days (RR1), including the duration of wet sequences (CWD) in southern Haiti (fig. 7c, 7d), while in southeastern Cuba, it is associated with an increase in mean rainfall intensity per rainy day (SDII) and heavy precipitation (R95p) (fig. 7b, 7e). This influence could be explained by the fact that La Nina brings wet conditions to the

Caribbean ( Klotzbach, 2011).

## 5.2   Impact of sea surface temperature anomalies (TSA, SST-Car)

The evolution of temperature anomalies in the South Atlantic (TSA average over 0-20S, 10E-30W) and Caribbean Sea (SST-Car SST average 14-16N, 65-85W) presented in suppl. Fig. 3 is marked by increasing warming in the Atlantic and Caribbean Sea over the period 1985-2015. In the Caribbean, warming has intensified over the past three decades in both seasons (DJF and

265 MAM) (suppl. fig.2). Thus, the +SST-Car phase in the Caribbean is associated with an increase in all extremes at the regional scale, apart from the mean rainfall intensity per wet day (SDII) (fig. 4). Similarly, on a local scale, considering all islands, it leads to an increase in the number of consecutive rainy days (CWD), as well as in the number of consecutive rainy days on the island of Hispaniola (Haiti and Santo Domingo) (fig. 6c, 6d). The increase in heavy precipitation (R95p) in Puerto Rico, southeast Cuba, and Haiti is also due to abnormally warm conditions + SST-Car in the Caribbean Sea(fig. 6b). These results

are in line with previous research on the influence of sea surface temperature on precipitation in the Caribbean, particularly in the Greater Antilles (Wang et al., 2007; Wang et al., 2008; Wu et al., 2011). Also, for the link between extreme precipitation and SSTs, it has been shown that the average rainfall intensity per wet day (SDII) averaged and heavy precipitation (R95p) over the Caribbean has a strong correlation with the warm phase of the Caribbean Sea (Peterson et al., 2002). In contrast, the positive +TSA phase (TSA averaged over 0-20S, 10E-30W), which corresponds to warmer sea surface temperatures (SST) in

the southern tropical Atlantic (TSA), is associated with a southward shift of the ITCZ (Philander et al., 1996) and a weakening of the southeasterly (SE) trade winds (Nobre and Shukla, 1996). This phase, which is also associated with higher precipitation in northeastern Brazil (Utida et al., 2019), influenced precipitation indices over the Greater Antilles, particularly in Puerto Rico and central Cuba, where it led to an increase in heavy precipitation (R95p) (fig.5b). On the other hand, in south-eastern Cuba and north-western Haiti, this phase was associated with an increase in rainfall intensity per wet day (SDII)(fig.6e). These results

are compared with those of the study by Utida et al. (2019), in which the influence of TSA on precipitation was assessed. The

results of this study show that the warm phase + TSA is associated with an increase in precipitation in southeastern Cuba. These findings are consistent with my own, namely that TSA influences southeastern Cuba.

## 6 Conclusions

This work provides a relevant analysis of the evolution of extreme precipitation and its link with global teleconnections over the Caribbean, particularly the Greater Antilles, over the period 1985-2015. Extreme precipitation indices (PRCPTOT, RR1, R95p, CWD, and CDD) defined by the World Meteorological Organization Expert Team on Climate Change Detection and Indices (ETCCDI) were calculated. Next, the links between large-scale SST oscillation indices (NAO, SOI, TSA, and SST-CAR) and extreme precipitation indices (PRCPTOT, RR1, R95p, CWD, and CDD) were evaluated and tested using Spearman's correlation coefficient. The results show that warming in the tropical South Atlantic (TSA), the Caribbean Sea (mean SST 14-16N, 65-85 W), and cooling in the eastern tropical Pacific (Niña) have positive effects on all extreme precipitation indices. Except for the number of rainy days (RR1) and rainy episodes (CWD), for which negative correlations were observed. However, the significant effects on extremes were greatest at the island scale in the Greater Antilles. For example, in southeastern Cuba and Puerto Rico, there was an increase in heavy precipitation (R95p) and average rainfall intensity per wet day (SDII) associated with the positive phase of the indices (SOI, TSA, and SST-Car), whereas in Jamaica and northern Haiti, there were only two indices (TSA and SST-Car). The number of rainy days (RR1) and the maximum duration of consecutive rainy days (CWD) showed a significant upward trend over southern Haiti and the Dominican Republic, in line with the positive phase of the Southern Oscillation (SOI) and warming east of the Caribbean Sea surface.

These results further improve our knowledge of the impact of certain global teleconnections on extreme precipitation in the Greater Antilles. They also highlight the most relevant teleconnection indices (SOI, SST-Car (average SST-Car SST 14-16N, 65–85 W), and TSA) to be considered as part of the impact study in the region, to limit damage to key economic sectors such as agriculture, biodiversity, health, and energy.

*Data availability.* The CHIRPS (Climate Hazards Group Infrared Precipitation with Stations data version 2) satellite data used in this study are available online at https://data.chc.ucsb.edu/products/CHIRPS-2.0/. NOAA OISST v2.1 data can also be accessed online at: (https://www.ncdc.noaa.gov/oisst/optimum-interpolation-sea-surface-temperature-oisst-v21).

*Author contributions.* Conceptualization: C.D, A.D, S.A; methodology: C.D, A.D, S.A; Original draft preparation: CD; review and editing: all the authors

*Competing interests.* The authors of this paper declare that they have no conflicts of interest.

*Financial support.* This research was carried out with the support of the following institutions: 1) ARTS PhD grant from the Institut de Recherche pour le Développement (IRD); 2) Antenor Firmin PhD grant from the French Embassy in Haiti; 3) CARIBACT International Joint Laboratory; 4) CLIMEXHA Project (Anticipation of Extreme CLIMATE events in HAITI for sustainable development

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

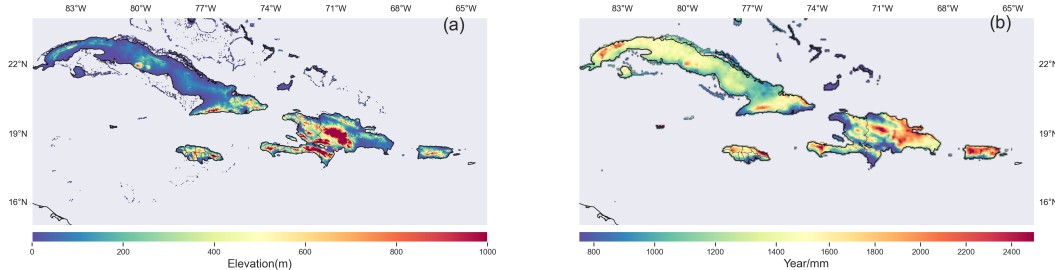

**Figure 1.** Location of study area. The figures show (a) the altitude of the four islands making up the Greater Antilles and (b) average annual rainfall.

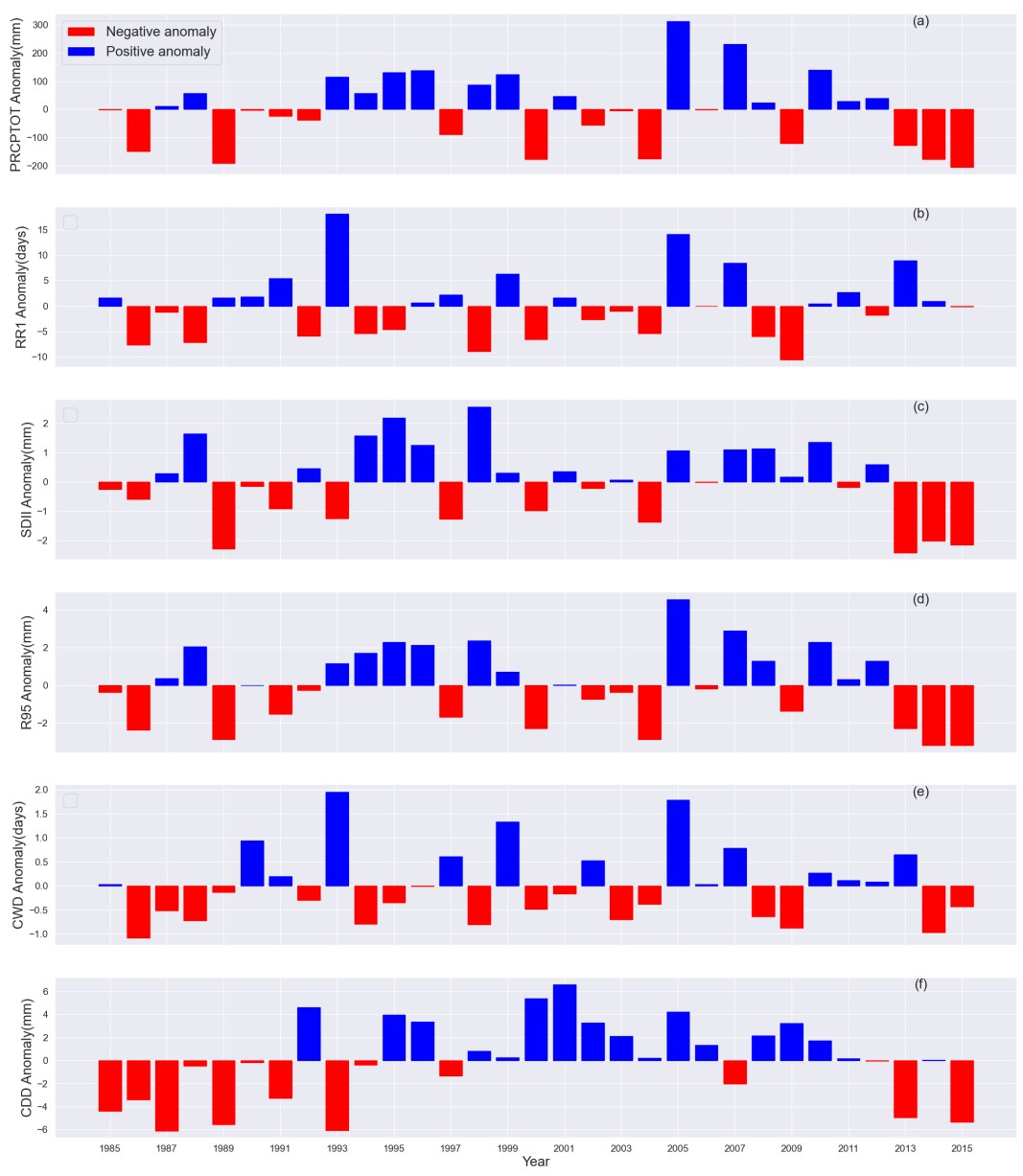

**Figure 2.** . Interannual variability of precipitation indices in the Greater Antilles over the period 1980-2015 with (a) PRCPTOT, (b) RR1, (c) SDII, (d) R95; (e) CWD; (f) CDD.

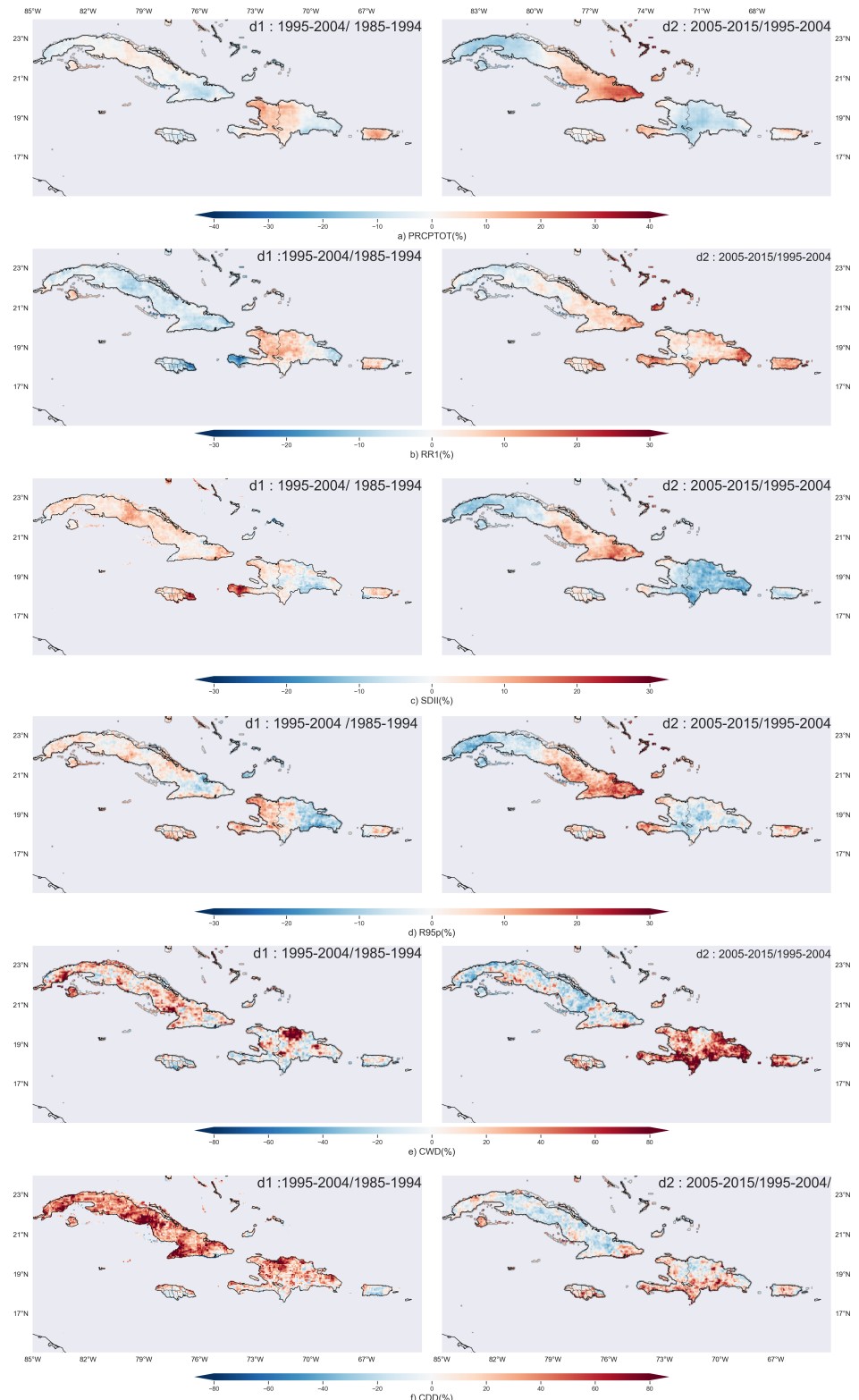

**Figure 3.** Annual change (%) between two consecutive decades of precipitation indices over the Greater Antilles (left: 1995-2004 compared to 1985-1994; right: 2005-2015 compared to 1995-2004 right): (a) PRCPTOT, (b) RR1, (c) SDII, (d) R95p, (e) CWD (f) CDD

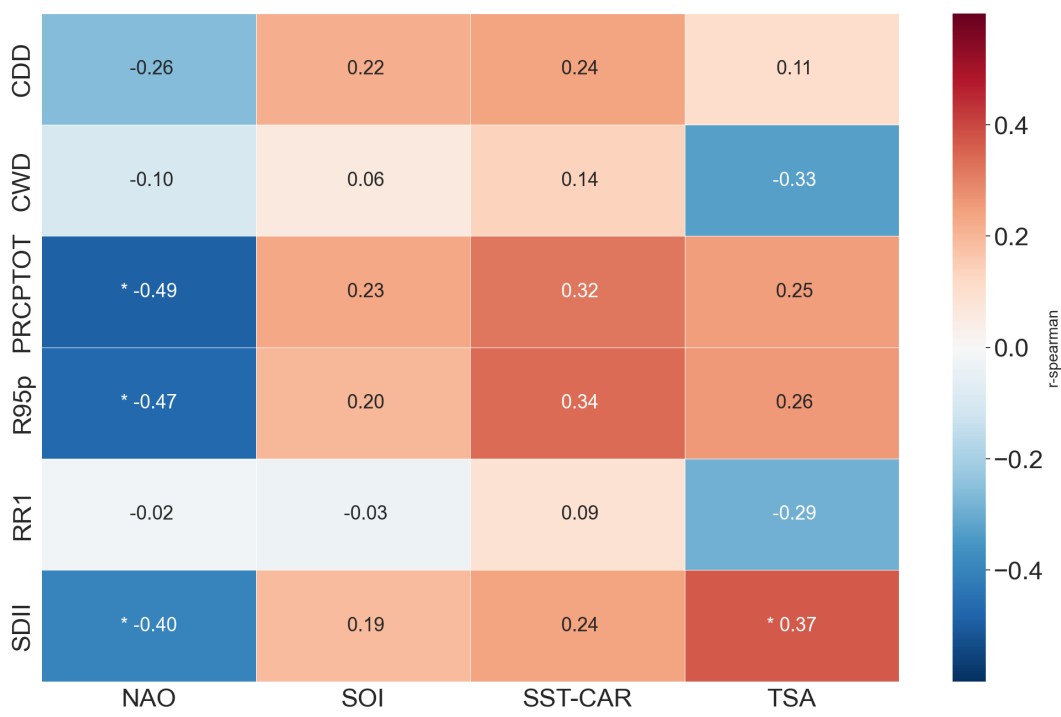

**Figure 4.** Correlation between precipitation indices and large-scale SST in the Greater Antilles (1985-2015). The values in the table are the correlation coefficients of large-scale SST with extremes. The indices on the abscissa are the precipitation extremes and those on the ordinate are the SST indices. The symbol (*) represents a statistically significant correlation at a threshold less than or equal to 0.05

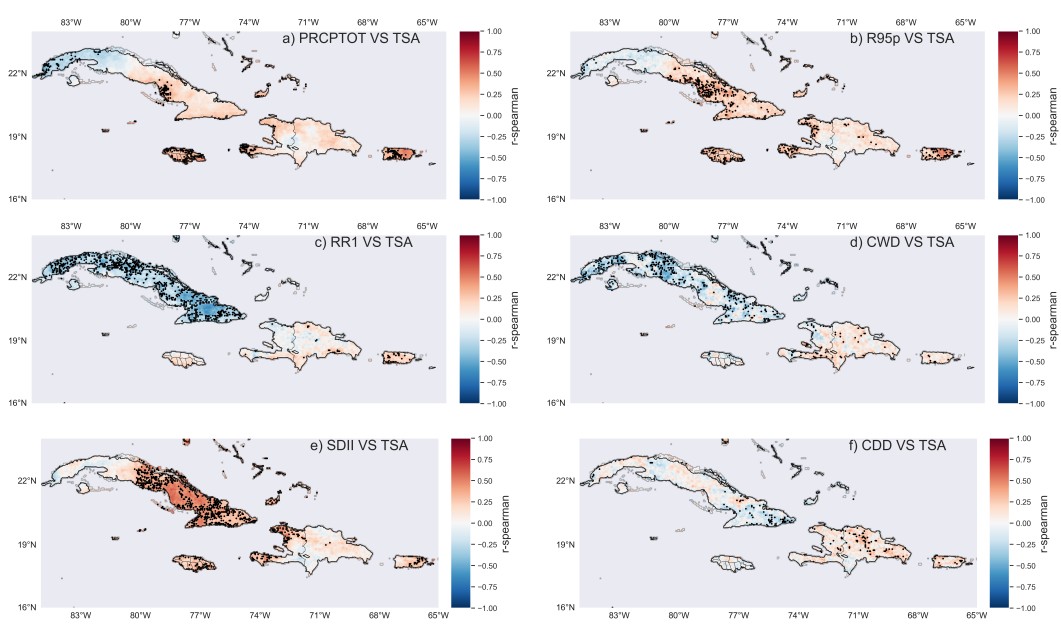

**Figure 5.** Correlation between precipitation indices and Tropical Southern Atlantic Index (TSA). The spatial correlation of extremes with TSA (SST average over 0-20S, 10E-30W) for this figure is presented in two columns; the first is realized with the indices: a) PRCPTOT Corr. TSA, b) RR1 Corr. TSA, c) SDII Corr. TSA and the second column with the indices: b) R95p Corr. TSA, d) CWD Corr. TSA, f) CDD Corr. TSA. Black dots represent areas where correlations are statistically significant at p ≤ 0.05.

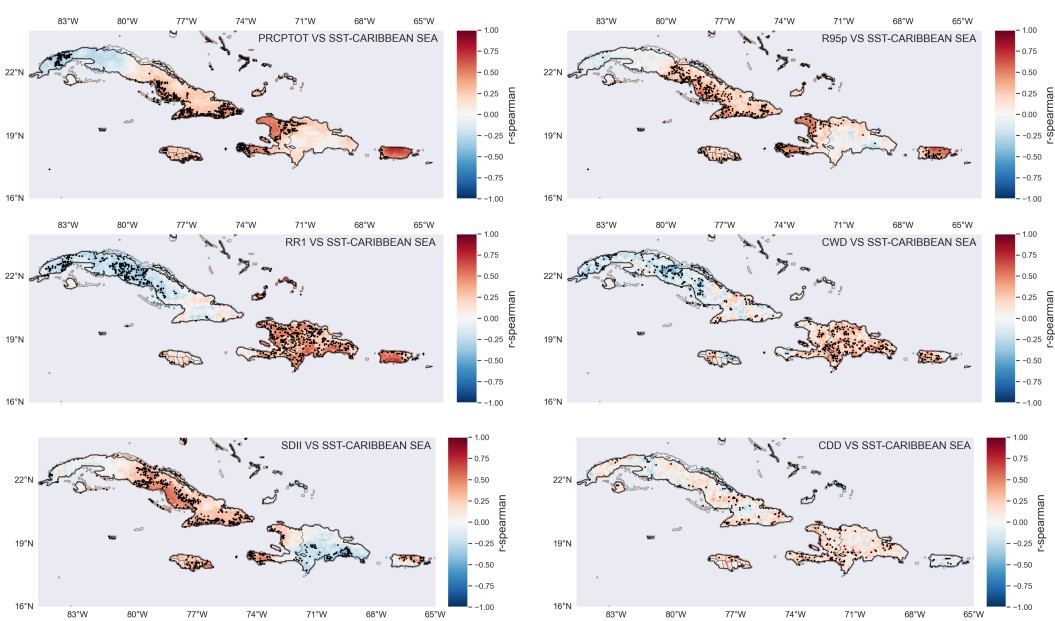

**Figure 6.** Correlation between precipitation indices and Caribbean Sea surface temperature. The spatial correlation of extremes with SST-Car(SST average 14-16N, 65-85W) for this figure is presented in two columns; the first is realized with the indices: a) PRCPTOT Corr. SST-Car, b) RR1 Corr. SST-Car, c) SDII Corr. SST-Car and the second column with the indices: b) R95p Corr. SST-Car, d) CWD Corr. SST-Car, f) CDD Corr. SST-Car. Black dots represent areas where correlations are statistically significant at p ≤ 0.05.

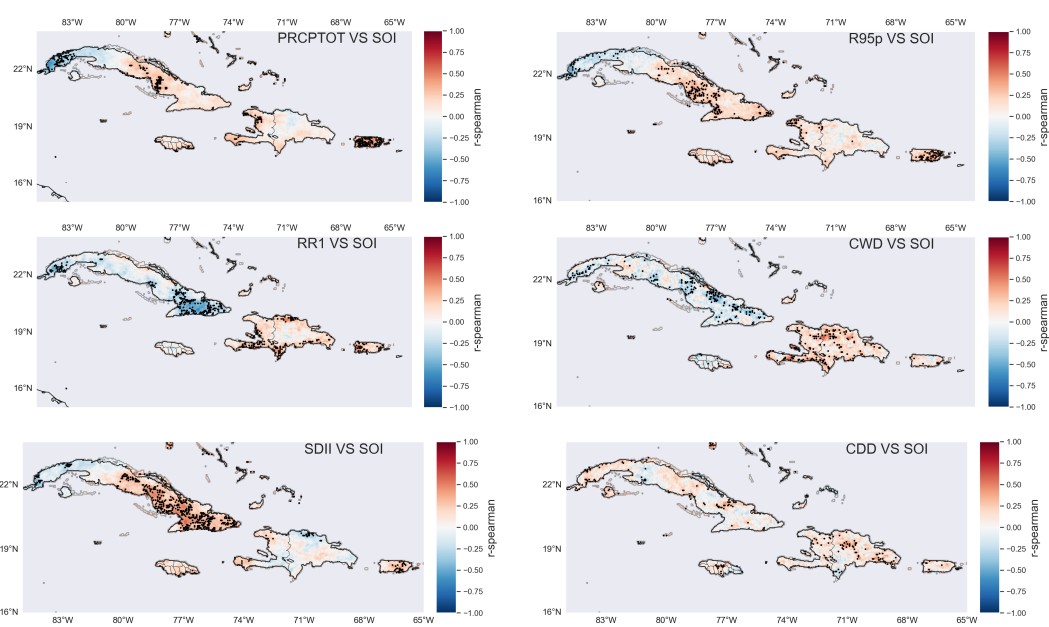

**Figure 7.** Correlation between precipitation indices and Southern Oscillation indices (SOI). The spatial correlation of extremes with SOI for this figure is presented in two columns; the first is realized with the indices: a) PRCPTOT Corr. SOI, b) RR1 Corr. SOI, c) SDII Corr. SOI and the second column with the indices: b) R95p Corr. SOI, d) CWD Corr. SOI, f) CDD Corr. SOI. Black dots represent areas where correlations are statistically significant at $p \leq 0.05$.

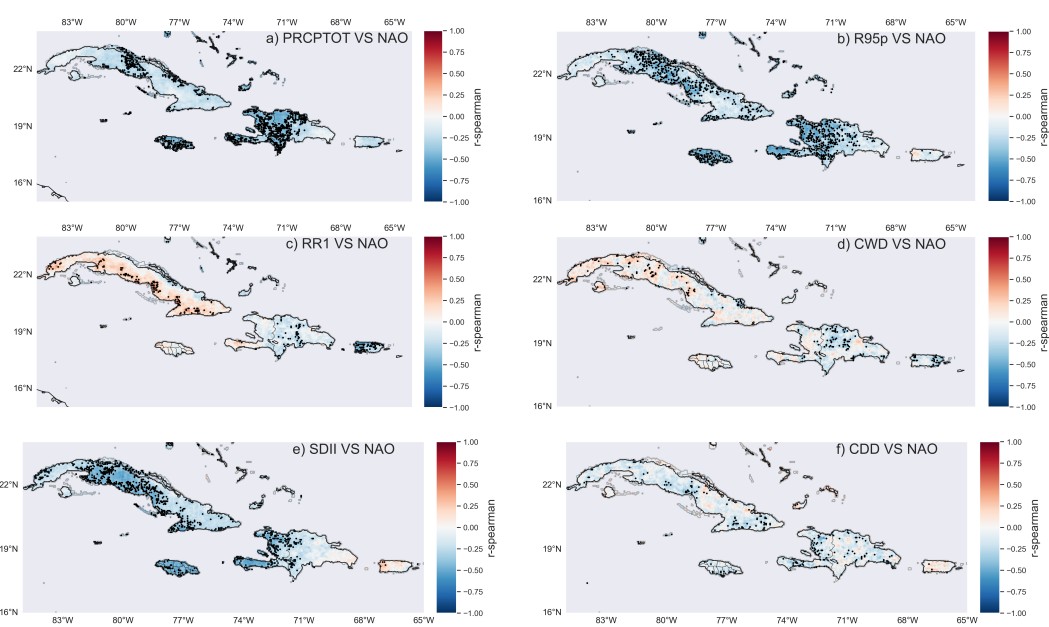

**Figure 8.** Correlation between precipitation indices and North Atlantic Oscillation Indices (NAO). The spatial correlation of extremes with NAO for this figure is presented in two columns; the first is realized with the indices: a) PRCPTOT Corr. NAO, b) RR1 Corr. NAO, c) SDII Corr. NAO and the second column with the indices: b) R95p Corr. NAO, d) CWD Corr. NAO, f) CDD Corr. NAO. Black dots represent areas where correlations are statistically significant at p ≤ 0.05

**Table 1.** Definition of extreme precipitation indices used in this study

| ID | Index Name | Indices definition | Units |
|---|---|---|---|
| RR1 | Total wet days index | Number of days with precipitation amount $\geq 1$ mm. Let RRij be the day daily precipitation amount on day i in period j. Count the number of days where: $RR_{ij} \geq 1$ mm. | days |
| PRCPTOT | Annual total precipitation on a wet day | Annual total rainfall precipitation on wet days. Let RRij be the daily precipitation amount on day i in period j. If I represent the number of days in I, then: PRCPTOTJ $= \sum_i^j RR_{ij}$ | mm |
| SDII | Simple daily rainfall intensity index | Simple daily rainfall intensity index: Let $RR_{ij}$ be the daily precipitation amount on a wet day, with RR> 1mm in period j. If W represents the number of wet days in j, then: $SDII_j = \frac{\sum_{w=1}^w RR_{wj}}{W}$ | mm/days |
| CWD | Consecutive wet days | Maximum number of consecutive days wet days. Let $RR_{ij}$ be the daily day precipitation amount on day i in period j. Count the largest number of consecutive days where: $RR_{ij} \geq 1$ mm | days |
| CDD | Consecutive dry days | Maximum number of consecutive dry days. Let $RR_{ij}$ be the daily day precipitation amount on day i in period j. Count the largest number of consecutive days where: $RR_{ij} < 1$ mm | days |
| R95p | Very wet days | The 95th percentile of daily precipitation events is the value above mm/-day which 5% of the daily precipitation events are found. | mm/day |