# Peer review of "Changes in extreme precipitation patterns over the Greater Antilles and teleconnection with large-scale sea surface temperature"

_Earth System Dynamics, 2024_

## Referee Comment (RC1)

Comments to the Author

Title: Changes in extreme precipitation patterns over the greater Caribbean and teleconnection with large-scale sea surface temperature.

Summary: This manuscript analyzes changes in extreme precipitation over the greater Caribbean and their correlation with large-scale sea surface temperature (SST) from 1985 to 2015. This is an important area of research as examining the impact of the key drivers on observed changes over the region, and particularly the links between extreme precipitation indices and large-scale sea surface indices are mandatory. While this type of analysis is valuable, the manuscript needs major revision. The paper should be reconsidered after major revision to better ground the conclusions and the presentation of the results.

General Comments:

1. Page 1, line 14: the phrase "the greater Caribbean" is used, What is this? I suggest the authors to clarify this.

2. Page 1, line 22: the phrase "Northern Oscillation Index (NAO)" is used, I highly recommend using "North Atlantic Oscillation" instead.

3. Page 3, lines 95-96 "They have a monthly rainfall cycle characterized by two peaks: the first in May and the second between September and November" This should be cited.

4. Page 3, lines 96-98: "North Atlantic anticyclone" better use "North Atlantic subtropical high (NASH)". Here adding more information about the Caribbean low-level Jet (CLLJ, Amador, 1998) and the Mid-Summer Drought will improve the text since both features are quite important in the climate of the region.

5. Page 3, lines 101-102, "The total annual precipitation in the Greater Antilles depends on land-sea interactions (breezes) and topography (fig.1a)" This should be cited.

6. Satellite data section. Due to complexities in estimating rainfall, for example, gridded products exhibit a very wide range of accuracy levels across regions worldwide. These products are developed at relatively high resolution or using sophisticated procedures, but even though, despite advances in estimating precipitation from satellite data, this option is limited by temporal sampling and algorithm errors that lead to advantages and limitations of each product. Why did the authors consider only using satellite data instead of other datasets? Why do the authors use only two products, when the use of more available products could lead to more robust results? According to the results, could the authors mention how these errors in satellite data could affect their results?

7. Page 4, lines 116-118: "TRMM 3B42v7 satellite product is used to calibrate and reduce the bias in the estimates". Could the authors be more specific? A bias correction method was applied. Why do the authors use this Tropical Rainfall Measuring Mission product instead of the integrated Multi-satellite retrievals for Global Precipitation Measurement (IMERG)?

8. Check every time CHIRPS is mentioned, first appeared as CHIRPSv2 and after as CHIRPS, please select one.

9. Page 4, lines 120-121 "evaluated over certain regions of the Americas, has demonstrated 121its ability to reproduce the mean climate as well as its capacity to estimate extreme precipitation events" Please add cites.

10. In general in the Satellite data section, I highly recommend adding the following reference "Centella-Artola A, Bezanilla-Morlot A, Taylor MA, Herrera DA, Martinez-Castro D, Gouirand I, Sierra-Lorenzo M, Vichot-Llano A, Stephenson T, Fonseca C, et al. Evaluation of Sixteen Gridded Precipitation Datasets over the Caribbean Region Using Gauge Observations. Atmosphere. 2020; 11(12):1334. https://doi.org/10.3390/atmos11121334". These will help to improve your research.

11. Why the NOAA DOISST (Daily Optimum Interpolation Sea Surface Temperature version 2.1) data is used?

12. Page 5, lines 150-155, I highly recommend to rewrite this paragraph.

13. More discussion should be added on why the use of Spearman correlation instead of Pearson for example.

14. Page 5, lines 156-162, I highly recommend to rewrite this paragraph.

15. Page 6, lines 166-169, What is the meaning of "whether the two variables are correlated or not"? If the trend or correlation is not statistically significant, then that means you cannot reject the null hypothesis (i.e., that there is no trend or correlation).

16. If the acronyms for the extremes were previously defined, please use them to reduce the text. Please check this.

17. Why the analysis is performed by decades instead of a full 30 years? what are the authors looking at with this?

18. Page 6, lines 176-179, I highly recommend to rewrite this sentence.

19. Page 6, lines 181-184, I highly recommend to rewrite this sentence. Besides, are these results related to a very active hurricane season, or are caused but something else?

20. Page 6, lines 185-196, I highly recommend to rewrite this paragraph.

21. Page 7, "Variations in extreme precipitation indices under the influence of variables such as NAO, SOI, TSA, and SST-CAR were analyzed over the Greater Antilles. The influences of large-scale variables were classified as positive, negative, positive, significant, negative, or significant, as shown in Figure 4." Could the authors explain the meaning of this?

22. Page 7, lines 209-210: based on what the authors made this comment? In Figure 4 I can not see this statement since significant correlations are barely seen.

23. Page 7, lines 210-213: this is impossible to see since the quality of the figures is low and one or two * did not make a difference.

24. I can not follow this discussion (figure 5-8) if the supplementary material is not available, besides the writing should be improved for better understanding.

25. Page 8, lines 249-253: please improve the writing.

26. The reference format should be revised.

27. All the figures' quality must be improved, Figure 3 revised the caption and the information on the figure does not coincide.

---

## Referee Comment (RC2)

**Review comments**

Manuscript title:

Changes in extreme precipitation patterns over the greater Caribbean and teleconnection with large-scale sea surface temperature

General comments:

I have thoroughly reviewed your manuscript and appreciate the effort and dedication that went into this research. Overall, the study explores the variation of precipitation characteristics and the impact from teleconnection patterns including regional SST anomaly and large-scale climate variability over the Caribbean region, particularly with the analysis conducted on a regional scale, which stands out as a significant highlight of your work.

However, there are areas that require substantial improvement before this manuscript can be considered for publication. Firstly, the writing needs enhancement to ensure clarity and coherence. Additionally, the visualization of data and results requires attention. Improving the readability and aesthetics of the figures and tables will greatly benefit the overall presentation and help convey your findings more effectively.

Given these considerations, I recommend major revisions to address these issues. Detailed comments are listed below.

1. Line 20: Would you please clarify 'a few large-scale SST index': to my understand, SOI and NAO are originally represented by specific patterns derived from pressures instead of SST.
2. Line 28, I would suggest to rewrite sentence 'In a ddition…'I assume it means the positive correlation between them are found, however, '+SOI significantly increases with RR1…' does not sounds the same meaning.
3. In the text, at many place the names for regions/countries are mentioned, I strongly suggest to add names for major locations in Figure 1.
4. Line 64-65: would you clarify the details about what 'climate extremes' were found have been increased over the region? Which I think this information is essential for building up the initiation of this research.
5. Line 74: I would suggest delete 'in'. I would suggest checking the references with brackets or not in the text, for example in Line 76, the backets is not necessary.
6. Line 75: would you clarify the meaning of AMO?
7. Line 81: what kind of further research is needed, I would suggest to introduce the research gap more detailed, like to what aspect or what kind of index is needed to be involved, etc.
8. Line84-86, 'In this context,….sea surface indices.' I would suggest rewriting this sentence to be clearer. Besides, here the term of 'sea surface indices' is used, instead of 'SST indices' (Line 20),

would you please consider using consistent terms to avoid any confusions.

9. Line 88: 'influence of SST indices on extreme precipitation', the indices are numbers used to represent the large scale climate phenomena, the impact should come from the phenomena, instead of the indices.

10. Line 98, what does 'They' refer to?

11. Line 99: Fig 1b shows the average annual rainfall, so it will not give any information on monthly maximum.

12. Line 99: The reference to figures is not correct. There is a lack in serial number for the sub plots in suppl.Fig2b, and this figure is seasonal decadal SST anomaly, not seasonal rainfall.

13. Line 104, 'because of topograpgy' I would suggest to describe the mechanism with more details.

14. Line 122: would you please clarify about the 'best result'? what kind of result, and best compared to what?

15. Line 124: would you please clarify 'heavy precipitation', does it refer to high intensity or large volume?

16. Line 133: I would suggest to rewrite the first sentence about the aim of developing extreme climate indices.

17. In Methodology, I suggest describing the calculation process of the indices, I wonder is it based on each grid or on regional average precipitation data.

18. Line 138: I would suggest delete '(6)'.

19. Table 1: would you describe the definition of indices with more detail? For example, it is unclear to define PRCPTOT with 'Annual total rainfall ≥ 1mm'.

20. Line 166, I would suggest to delete 'that is, whether the two variables are really correlated or not,' just want to clarify that 0,05 is a level of significance, threshold less than 0,05 means higher confidence in significant level.

21. Please clarify the meaning of n in equation 3.

22. Line 174: misspelling PRCPTOT

23. Line 177: Could you please clarify why 'a decline in total annual precipitation in the first decade' is concluded? I assume that the bar plot shows annual mean anomaly of extreme precipitation indices, and I would suggest to apply a trend analysis to see if it exists any increase/decrease.

24. Line 185: I would suggest use 'percentage' instead of %. Please consider to give each subplot a number to make it easier and clearer to indicate to specific ones. In Figure 3, I personally have curious in the change not only between decade1-2 and decade 2-3, but also 1-3. Moreover, there seems no discussion in text about the difference between each decades, I would suggest to reconsider the methodology and data visualization of this figure.

25. Line 204, To my understanding in this study there is no calculations in difference phase of climate indices (the correlation analysis should be done with continues climate indices instead of separating them into different phases), if so, please rewrite the sentence. Similar statement can be found in Line 243, 'positive phase of +NAO….', also in Discussion section, eg, Line 262, Line

266. Please check an example of analysing the different phase of climate indices:

26. Line 211: the * and ** symbol needs to be defined better, for example, * could be used to represent the regions with less than 50% of the area significant, if larger, than marked as **.

27. In supplement Table 1, should be INDEX-NAO, Haitu should be Haiti?

28. Line 224: there exists no Suppl.Table 2.

29. Line 254: the effect of sea surface anomalies. I would suggest to consider using the term 'impact or influence' other than 'effect'.

30. Line 317: '…and cooling in the eastern Pacific (La Niña) have positive and significant effects on extreme precipitation indices.' Would you please clarify the basis/evidence for this conclusion?

---

## Author Comment (AC3)

**Changes in extreme precipitation patterns over the greater Caribbean and teleconnection with large-scale sea surface temperature**

**Manuscript No. ESD-2024-15**

Carlo Destouches, Arona Diedhiou, Sandrine Anquetin, Benoit Hingray, Armand Pierre, Dominique Boisson, and Adermus Joseph

**Reply to reviewer #1**

We sincerely thank reviewer #1 for his thorough review and insightful comments. Below are our responses to the comments. Responses are shown in blue, comments in black.

1. Page 1, line 14: the phrase "the greater Caribbean" is used, What is this? I suggest the authors to clarify this.

   The Greater Caribbean comprises all the states and territories with coastlines on the Caribbean Sea. These include the Greater Antilles and the Lesser Antilles. The term "la grande cariabe" has therefore been used to present the region from which the study area has been extracted. However, in line 14, the term "La grande Caraibe" could mean that the work was carried out over the whole region. To clarify, a correction has been made in the text by adding the following: "the greater Caribbean, particularly the Greater Antilles".

2. Page 1, line 22: the phrase "Northern Oscillation Index (NAO)" is used, I highly recommend using "North Atlantic Oscillation" instead.

   This correction will be taken into account in the revised version of the document.

3. Page 3, lines 95-96 "They have a monthly rainfall cycle characterized by two peaks: the first in May and the second between September and November" This should be cited.

   Chen and Taylor(2002), this citation will be added to the revised version of the paper.

4. Page 3, lines 96-98: "North Atlantic anticyclone" better use "North Atlantic subtropical high (NASH)". Here adding more information about the Caribbean low-level Jet (CLLJ, Amador, 1998) and the Mid-Summer Drought will improve the text since both features are quite important in the climate of the region.

   This paragraph will be added to the revised version:

   The term North Atlantic subtropical high (NASH) has been substituted for North Atlantic anticyclone in the text. Also, to improve content on the low-level jet in carabids, we've added the paragraph below:

The low-level jet (CLLJ), which is characterized by two maximums; the first in January and the second in July (Amador, 1998; Cook and Vizy, 2010), plays a very important role in transporting moisture to the Caribbean (Mo et al., 2005). Also, the July peak is associated with a short dry season (Wang and Lee 2007).

5. Page 3, lines 101-102, "The total annual precipitation in the Greater Antilles depends on land-sea interactions (breezes) and topography (fig.1a)" This should be cited.

   Two citations will be added to the revised version of the paper:  Cantet(2007)  et Moron et al. (2015)

6. Satellite data section. Due to complexities in estimating rainfall, for example, gridded products exhibit a very wide range of accuracy levels across regions worldwide. These products are developed at relatively high resolution or using sophisticated procedures, but even though, despite advances in estimating precipitation from satellite data, this option is limited by temporal sampling and algorithm errors that lead to advantages and limitations of each product. Why did the authors consider only using satellite data instead of other datasets? Why do the authors use only two products, when the use of more available products could lead to more robust results? According to the results, could the authors mention how these errors in satellite data could affect their results?

   Thanks for the interesting questions. In fact, at first, we wanted to use data observed on the ground. We used NOAA's Global Surface Summary of the Day (GSOD) databases. However, given the high percentage of missing data in the time series, we opted for satellite data. As you mentioned in the comments, satellite data have limitations due to algorithm errors during estimation. So, before using these products, it is important to evaluate their performance. In the area, the study by Bathelemy et al (2022), in which several satellite products were evaluated, revealed that CHIRPS satellite data reproduce rainfall seasonality very well and perform well in estimating heavy rainfall in the Caribbean, particularly the Greater Antilles. Although other satellite products were used, such as MSWEP, for which good performance was also observed, CHIRPs was the most judicious choice because of its high resolution (5kmx5km) compared with other satellite products.

   It should also be noted that Chirps underestimates rainfall in dry seasons and overestimates rainfall in wet seasons (Fig.7, Bathelemy et al. (2022)). Although these discrepancies are not significant, when interpreting the results, we consider them to be borderline.

7. Page 4, lines 116-118: "TRMM 3B42v7 satellite product is used to calibrate and reduce the bias in the estimates". Could the authors be more specific? A bias correction method was applied. Why do the authors use this Tropical Rainfall Measuring Mission product instead of the integrated Multi-satellite retrievals for Global Precipitation Measurement (IMERG)?

The work of Funk et al. (2015) revealed that the TRMM 3B42v7 satellite product was used to calibrate the Chirps data. As in the given section of this study, a description of the CHirps satellite product was made, to this effect, we added the TRMM 3B42v7 satellite product to complete this description by referring to the work of Funk et al. (2015). However, in relation to the question on the IMERG satellite product, this is an interesting one. We wonder whether this one concerns us directly in the context of this study.

8. Check every time CHIRPS is mentioned, first appeared as CHIRPSv2 and after as CHIRPS, please select one.

   In the revised version, CHIRPS will be replaced in the text by CHIRPSv2.

9. Page 4, lines 120-121 "evaluated over certain regions of the Americas, has demonstrated 121its ability to reproduce the mean climate as well as its capacity to estimate extreme precipitation events" Please add cites.

   Dans la version révisée, on ajoutera cette citation : Rivera et al.(2019)

10. In general in the Satellite data section, I highly recommend adding the following reference "Centella-Artola A, Bezanilla-Morlot A, Taylor MA, Herrera DA, Martinez-Castro D, Gouirand I, Sierra-Lorenzo M, Vichot-Llano A, Stephenson T, Fonseca C, et al. Evaluation of Sixteen Gridded Precipitation Datasets over the Caribbean Region Using Gauge Observations. Atmosphere. 2020; 11(12):1334. https://doi.org/10.3390/atmos11121334". These will help to improve your research.

   In the revised version, we will add this citation.

11. Why the NOAA DOISST (Daily Optimum Interpolation Sea Surface Temperature version 2.1) data is used?

   The NOAA DOISST choice is justified in the above graph:

   Sea surface temperatures (SST) are very important for monitoring and assessing climate change (IPCC 2013). They can be derived either from observations from floating or moored buoys (Smith et al. 1996), from satellite observations (Merchant et al., 2014), or from a mixture (in situ + satellite) (HadSST, Rayner et al., 2003; Reynold et al., 2007, DOISST). Given that NOAA DOISST(Reynold et al., 2007, Huang et al., 2021 ) has been widely used for climate assessment and monitoring, notably as part of the reanalysis of the NOAA/NCEP climate prediction system(Saha et al., 2010). Also, work by Huang et al.(2021a) has revealed that NOAA DOISST performs well in terms of bias compared with buoy and Argo observations, as well as with the eight SST products.

12. Page 5, lines 150-155, I highly recommend to rewrite this paragraph.

This paragraph will be rewritten in the revised version with more detail on the influence of Atlantic (Pacific) Ocean SSTs on precipitation in the Caribbean. Also, the term precipitation variability will preferably be replaced by precipitation has been influenced by SSTs.

13. More discussion should be added on why the use of Spearman correlation instead of Pearson for example.

A few lines of discussion I'd like to add to the revised version:

Analysis of the relationship between two variables is often of great interest for data analysis in research. It generally consists in characterizing the form and intensity of the link (relationship) between variables by means of a correlation coefficient. For two variables, X and Y, this coefficient is interpreted as : i)linear linkage, the correlation coefficient is positive when X and Y values change in the same direction, i.e., an increase in X leads to an increase in Y; ii)linear linkage, the correlation coefficient is negative when X and Y values change in the opposite direction, i.e., an increase in X leads to a decrease in Y (or vice versa); iii) non-linear monotonic linkage, the correlation coefficient is positive when X and Y change in the same direction as in (i), but with a small slope (Lewis-Beck, 1995; Sheskin, 2007; Gibbons).

In the literature, Pearson's and Spearman's correlation coefficients are often the most widely used to measure the strength or degree of linkage between two variables. In this study, we used Spearman's non-parametric rank correlation coefficient (rho) to assess the interannual link between extreme precipitation over indices and global SSTs indices. This non-parametric method was chosen because it does not require a normal distribution for the variables. Also, Spearman outperforms Pearson's linear coefficient in the case of outliers. On the other hand, linear trends can be detected using Pearson or Spearman tests, but the latter is preferable for monotonic non-linear relationships (Gauthier, 2001; Von Storch and Zwiers, 1999).

14. Page 5, lines 156-162, I highly recommend to rewrite this paragraph.

The few contents #13 will allow me to improve this paragraph in the revised version.

15. Page 6, lines 166-169, What is the meaning of "whether the two variables are correlated or not"? If the trend or correlation is not statistically significant, then that means you cannot reject the null hypothesis (i.e., that there is no trend or correlation).

Two variables are correlated when an increase in one variable leads to an increase in the other, or a decrease(increase) in one variable leads to an increase(decrease) in the other. However, when analyzing data, statistical tests are available in the literature to determine whether there is indeed a statistically significant correlation (or not) between two variables. The correlation is statistically significant if the null hypothesis is not verified. On the other hand, if the null hypothesis is verified, the correlation is not statistically significant.

16. If the acronyms for the extremes were previously defined, please use them to reduce the text. Please check this.

This correction will be considered in the revised version.

17. Page 6, lines 176-179, I highly recommend to rewrite this sentence.

Thank you, in the revised version this correction will be considered.

18. Page 6, lines 181-184, I highly recommend to rewrite this sentence. Besides, are these results related to a very active hurricane season, or are caused but something else?

For lines 181-184, this correction will be considered in the revised version. For this question relating to cyclonic seasons, unfortunately in this study, the results do not take cyclonic seasons into account.

19. Page 6, lines 185-196, I highly recommend to rewrite this paragraph

Thank you, this correction will be considered in the revised version.

20. Page 7, "Variations in extreme precipitation indices under the influence of variables such as NAO, SOI, TSA, and SST-CAR were analyzed over the Greater Antilles. The influences of large-scale variables were classified as positive, negative, positive, significant, negative, or significant, as shown in Figure 4." Could the authors explain the meaning of this?

The term influence used here refers to the correlation between SSTs and precipitation indices. As explained in answer #13, the variables can be either varied in the same direction or in the opposite direction. Hence the need to associate signs (positive, negative) to characterize the direction of variation. On the other hand, once the direction of the correlation has been determined, a statistical test is needed to determine the significance of the sign. In other words, whether this positive (negative) correlation is statistically significant or not.

21. Page 7, lines 209-210: based on what the authors made this comment? In Figure 4 I can not see this statement since significant correlations are barely seen.

Significant correlations are the correlation coefficients in Figure 4 with the symbols *. This symbol is placed to the left of the correlation coefficient.

22. Page 7, lines 210-213: this is impossible to see since the quality of the figures is low and one or two * did not make a difference.

Thank you for your comments, the quality of the figures will be improved. As these figures have been produced on Python, we will use a python package(searbon) to improve the quality of the figures.

23. I can not follow this discussion (figure 5-8) if the supplementary material is not available, besides the writing should be improved for better understanding.

There is no supplementary material for this section. However, we promise that the writing will be improved to facilitate comprehension in the revised version.

24. Page 8, lines 249-253: please improve the writing.

This correction will be considered in the revised version.

25. The reference format should be revised.

This correction will be considered using the format proposed by ESD in the revised version.

26. All the figures' quality must be improved, Figure 3 revised the caption and the information on the figure does not coincide.

We're going to use another python package (seaborn) to improve the quality of the figures.

---

## Author Comment (AC4)

**Changes in extreme precipitation patterns over the greater Caribbean and teleconnection with large-scale sea surface temperature**

**Manuscript No. ESD-2024-15**

Carlo Destouches, Arona Diedhiou, Sandrine Anquetin, Benoit Hingray, Armand Pierre, Dominique Boisson, and Adermus Joseph

**Reply to reviewer #2**

We sincerely thank reviewer #2 for his thorough review and insightful comments. Below are our responses to the comments. Responses are shown in blue, comments in black.

1. Line 20: Would you please clarify 'a few large-scale SST index': to my understand, SOI and NAO are originally represented by specific patterns derived from pressures instead of SST.

   Thanks for the comment, indeed, SST indices should be specified i.e., whether they are derived from pressure variation or sea surface temperature. Since this line concerns the abstract, we wonder if these details should not be added in the methodology instead of in the abstract.

2. Line 28, I would suggest to rewrite sentence 'In a ddition…'I assume it means the positive correlation between them are found, however, '+SOI significantly increases with RR1…' does not sounds the same meaning.

   In fact, significant positive correlations were obtained. These positive correlations were also expressed in this sentence, which would mean that Caribbean Sea warming and the positive +SOI phase are associated with an increase in the indices (RR1, CWD). This increase is statistically significant, since a statistical test has been applied to this effect. This sentence will be rewritten to make it clearer.

3. In the text, at many place the names for regions/countries are mentioned, I strongly suggest to add names for major locations in Figure 1.

   This correction will be considered in the revised version.

4. Line 64-65: would you clarify the details about what 'climate extremes' were found have been increased over the region? Which I think this information is essential for building up the initiation of this research.

Thanks, that's a very interesting question. We'll add these details in the revised version. Climate extremes are extreme rainfall, drought and wet spells.

5.  Line 74: I would suggest delete 'in'. I would suggest checking the references with brackets or not in the text, for example in Line 76, the backets is not necessary.

    It's been noted and will be included in the revised version.

6.  Line 75: would you clarify the meaning of AMO?

    This precision will be added in the revised version.

7.  Line 81: what kind of further research is needed, I would suggest to introduce the research gap more detailed, like to what aspect or what kind of index is needed to be involved, etc.

    Thank you for your comment. These details will be added in the revised version.

8.  Line84-86, 'In this context,….sea surface indices.' I would suggest rewriting this sentence to be clearer. Besides, here the term of 'sea surface indices' is used, instead of 'SST indices' (Line 20), would you please consider using consistent terms to avoid any confusions.

    In fact, the term "sea surface index" does not include temperature. We will correct this to avoid any confusion.

9.  Line 88: 'influence of SST indices on extreme precipitation', the indices are numbers used to represent the large scale climate phenomena, the impact should come from the phenomena, instead of the indices.

    Thank you for your comment, but I didn't quite understand what you were suggesting.

10. Line 98, what does 'They' refer to?

    Thanks, it's the climatology of monthly rainfall that's influenced....We'll make the correction in the revised version.

11.  Line 99: Fig 1b shows the average annual rainfall, so it will not give any information on monthly maximum.

    Thank you, there was an error; the reference is in the figures (sup.fig 1b) in the supplementary file.

12. Line 99: The reference to figures is not correct. There is a lack in serial number for the sub plots in suppl.Fig2b, and this figure is seasonal decadal SST anomaly, not seasonal rainfall.

    Thanks, indeed, the reference is rather suppl.Fig1b.

13. Line 104, 'because of topograpgy' I would suggest to describe the mechanism with more details.

More details will be added in the revised version.

14. Line 122: would you please clarify about the 'best result'? what kind of result, and best compared to what?

This line presents the results of a study in which a better performance of the Chrips satellite product was used to reproduce in situ data.

15. Line 124: would you please clarify 'heavy precipitation', does it refer to high intensity or large volume?

Heavy rainfall was defined on the basis of a relative threshold (percentile) over a given period.

16. Line 133: I would suggest to rewrite the first sentence about the aim of developing extreme climate indices.

Noted. This correction will be added in the revised version.

17. In Methodology, I suggest describing the calculation process of the indices, I wonder is it based on each grid or on regional average precipitation data.

Table 1 of precipitation indices is not sufficient for this question. In this table, a definition has been associated with each index.

18. Line 138: I would suggest delete '(6)'.

Noted.

19. Table 1: would you describe the definition of indices with more detail? For example, it is unclearto define PRCPTOT with 'Annual total rainfall ≥ 1mm'.

Thank you, more details will be added in the revised version.

20. Line 166, I would suggest to delete 'that is, whether the two variables are really correlated or not,'just want to clarify that 0,05 is a level of significance, threshold less than 0,05 means higher confidence in significant level.

Thank you, noted. This correction will be added in the revised version.

21. Please clarify the meaning of n in equation 3.

Noted. This correction will be added in the revised version.

22. Line 174: misspelling PRCPTOT.

Noted. This correction will be added in the revised version.

23. Line 177: Could you please clarify why 'a decline in total annual precipitation in the first decade'is concluded? I assume that the bar plot shows annual mean anomaly of extreme precipitation indices, and I would suggest to apply a trend analysis to see if it exists any increase/decrease.

An initial explanation for this drop could be based on the number of rainy days and average rainfall intensity over the year. This is consistent with the fact that total annual precipitation depends on the number of rainy days and average rainfall intensity respectively. This explanation is given in the text (line 190).

24. Line 185: I would suggest use 'percentage' instead of %. Please consider giving each subplot a number to make it easier and clearer to indicate to specific ones. In Figure 3, I personally have curious in the change not only between decade1-2 and decade 2-3, but also 1-3. Moreover, there seems no discussion in text about the difference between each decade, I would suggest to reconsider the methodology and data visualization of this figure.

Thank you for your suggestions. Regarding the suggested numbers, letters have been assigned to each figure group. For the difference between the decades, in the revised version, we'll be adding a lot more discussion.

25. Line 204, To my understanding in this study there is no calculations in difference phase of climate indices (the correlation analysis should be done with continues climate indices instead of separating them into different phases), if so, please rewrite the sentence. Similar statement can be found in Line 243, 'positive phase of +NAO....', also in Discussion section, eg, Line 262, Line 266. Please check an example of analysing the different phase of climate indices:

To be more explicit, we looked at whether there is a link between precipitation and SST indices, in other words, whether an increase (decrease) in one SST index leads to an increase in the other (if we refer to the definition of a link between two variables). To calculate this correlation, we used the data anomalies of the two variables. For this purpose, we used the term phase to refer to the signs of anomalies.

26. Line 211: the * and ** symbol needs to be defined better, for example, * could be used to represent the regions with less than 50% of the area significant, if larger, than marked as **.

Thank you for this suggestion, it will be considered in the revised version.

27. In supplement Table 1, should be INDEX-NAO, Haitu should be Haiti?

Thank you, we will make this correction in the revised version.

28. Line 224: there exists no Suppl.Table 2.

Thank you, there was an error. It's Suppl.Table 1, this correction will be added in the revised version.

29. Line 254: the effect of sea surface anomalies. I would suggest to consider using the term 'impactor influence' other than 'effect'.

Noted. This correction added in the revised version.

30. Line 317: '…and cooling in the eastern Pacific (La Niña) have positive and significant effects on extreme precipitation indices.' Would you please clarify the basis/evidence for this conclusion?

This clarification will be added in the revised version.

---

## Referee Report (RR1)

Comments to the Author

Title: Changes in extreme precipitation patterns over the greater Caribbean and teleconnection with large-scale sea surface temperature.

Summary: This manuscript analyzes changes in extreme precipitation over the greater Caribbean and their correlation with large-scale sea surface temperature (SST) from 1985 to 2015. This is an important area of research as examining the impact of the key drivers on observed changes over the region, and particularly the links between extreme precipitation indices and large-scale sea surface indices are mandatory. The authors has taken into account all the suggestions. However, since some issues remain, the manuscript needs minor revision. The paper should be reconsidered after minor revision.

General Comments:

1. There are some typo mistakes. Please check.

2. Again the CHIRPS dataset, sometimes appeared as chirps, Chirps, or CHIRPS-v2, please be consistent.

3. The acronym NASH is not defined.

4. Figures need to be improved since the axis can barely be seen, please improve quality.

---

## Editor Decision (ED1)

Dear Carlo and co-authors,

Thank you for the submission of your interesting manuscript "Changes in extreme precipitation patterns over the greater Caribbean and teleconnection with large-scale sea surface temperature".

As you know, two reviewers have now provided detailed reviews, which you have replied in thoughtful detail to. Both reviewers recommended major revisions and therefore I would like to invite you to submit a revised version of your manuscript.

Would you please also provide an 'author's reply' to the reviewers (feel free to use the same words that you used in what you have already uploaded). Please can you also include a track changes document between the old manuscript and the new one (you can include this as part of your 'author's reply').

In addition to the suggestions from the reviewers, I would like the authors to consider the following comments:

(a) Can the authors provide a statement about the data and code availability?
(b) Can the authors clarify in the revised manuscript why this specific precipitation dataset was used for this analysis and discuss the robustness of their findings?
(c) All terms used in the abstract (such as SST) need to be clarified. The abstract should be fully understandable without the reader having to read the manuscript.
(d) All figures should be readable and follow the journal's standards (for e.g. width not less than 8cm, minimum resolution of 300 dpi, see https://www.earth-system-dynamics.net/submission.html#figurestables ).
(e) Statements about correlation and their statistical significance should be carefully reviewed. If a result is not significant, you fail to reject the null hypothesis. Therefore, while one can say that a correlation is statistically significant, one cannot conclude that a lack of significance means no correlation.
(f) Although your manuscript will undergo a copy editing at the final stage, there are sentences in your manuscript which one cannot follow due to the issues of English. I am not saying the English must be grammatically perfect, but to a level that it can clearly convey the scientific reasoning and findings.

I look forward to seeing the next version of your manuscript which I will then send out for further review to either the previous reviewers (if they agree) or new reviewers.

Best regards,

Dr. Anaïs Couasnon